# Safety, Efficacy, and Outcomes of N-Butyl Cyanoacrylate Glue Injection through the Endoscopic or Radiologic Route for Variceal Gastrointestinal Bleeding: A Systematic Review and Meta-Analysis

**DOI:** 10.3390/jcm10112298

**Published:** 2021-05-25

**Authors:** Olivier Chevallier, Kévin Guillen, Pierre-Olivier Comby, Thomas Mouillot, Nicolas Falvo, Marc Bardou, Marco Midulla, Ludwig-Serge Aho-Glélé, Romaric Loffroy

**Affiliations:** 1Department of Vascular and Interventional Radiology, Image-Guided Therapy Center, ImViA Laboratory-EA 7535, François-Mitterrand University Hospital, 14 Rue Paul Gaffarel, BP 77908, 21079 Dijon, France; olivier.chevallier@chu-dijon.fr (O.C.); kguillen@hotmail.fr (K.G.); nicolas.falvo@chu-dijon.fr (N.F.); marco.midulla@chu-dijon.fr (M.M.); 2Department of Neuroradiology and Emergency Radiology, François-Mitterrand University Hospital, 14 Rue Paul Gaffarel, BP 77908, 21079 Dijon, France; pierre-olivier.comby@chu-dijon.fr; 3Department of Gastroenterology and Hepatology, François-Mitterrand University Hospital, 14 Rue Paul Gaffarel, BP 77908, 21079 Dijon, France; thomas.mouillot@chu-dijon.fr (T.M.); marc.bardou@chu-dijon.fr (M.B.); 4Department of Biostatistics and Epidemiology, François-Mitterrand University Hospital, 14 Rue Paul, Gaffarel, BP 77908, 21079 Dijon, France; ludwig.aho@chu-dijon.fr

**Keywords:** gastrointestinal hemorrhage, variceal bleeding, cyanoacrylates, embolization, portal hypertension, cirrhosis

## Abstract

We performed a systematic review and meta-analysis of published studies to assess the efficacy, safety, and outcomes of N-butyl cyanoacrylate (NBCA) injection for the treatment of variceal gastrointestinal bleeding (GIB). The MEDLINE/PubMed, EMBASE, and SCOPUS databases were searched for English-language studies published from January 1980 to December 2019 and including patients who had injection of NBCA for variceal GIB. Two independent reviewers extracted and evaluated the data from eligible studies. Exclusion criteria were sample size < 5, article reporting the use of NBCA with other embolic agents, no extractable data, and duplicate reports. NBCA was injected during endoscopy in 42 studies and through a direct percutaneous approach for stomal varices in 1 study. The study’s endpoints were: Technical success, 30-day rebleeding, and 30-day overall and major complications. The estimated overall rates were computed with 95% confidence intervals, based on each study rate, weighted by the number of patients involved in each study. In total, 43 studies with 3484 patients were included. The technical success rate was 94.1% (95% CI: 91.6–96.1%), the 30-day rebleeding rate was 24.2% (18.9–29.9%), and 30-day overall and major complications occurred in 15.9% (11.2–21.3%) and 5.3% (3.3–7.8%) of patients, respectively. For treating variceal GIB, NBCA injection is a safe and effective method that demonstrates high technical success rate and very low major complication rate.

## 1. Introduction

Acute variceal bleeding is a life-threatening complication of portal hypertension and the cause of death in about one-third of patients with liver cirrhosis [1]. Other less common conditions might result in portal hypertension and variceal gastrointestinal bleeding (GIB), such as splenic vein thrombosis, hepatic sinusoidal obstruction syndrome, and primary biliary cirrhosis. Gastroesophageal varices are the most common type and are responsible for upper GIB (UGIB), with esophageal varices being the most frequent [2]. Furthermore, ectopic varices may develop anywhere along the digestive tract (duodenum, small bowel, colon, rectum, stomies) in the presence of portal hypertension and may cause lower GIB (LGIB) [3,4,5,6]. To ensure an effective therapy, a multidisciplinary approach including gastroenterologists, hepatologists, critical care physicians, surgeons, and interventional radiologists is mandatory. Practice guidelines for the management of variceal GIB in cirrhotic patients are recommended by the American Association for the Study of Liver Disease (AASLD) [7] and the European Association for the Study of the Liver (EASL) [8]. Therapies should be chosen according to the different stages of cirrhosis and portal hypertension. Acute variceal GIB can be managed through various methods used alone or in combination, including endoscopic therapy, the use of vasoactive drugs, balloon tamponade, endoscopically self-expandable metal stent placement, esophageal transaction, transjugular intrahepatic portosystemic shunt (TIPS) with or without varices embolization, balloon-occluded retrograde transvenous obliteration (BRTO), and varices embolization through transsplenic route. Endoscopic variceal ligation (EVL) remains the main endoscopic therapeutic option [7,8,9]. However, in the presence of massive bleeding or in cases of non-esophageal varices, such as cardiofundal gastric varices, EVL can be challenging and ineffective. In these situations, sclerotherapy might be more appropriate [10,11,12]. Injection of N-butyl cyanoacrylate (NBCA) has been used for varices obturation and has demonstrated good results. For gastric varices, NBCA injection may be as effective as EVL for initial hemostasis with a lower rebleeding rate [13]. Due to the heterogeneous bleeding location and anatomy of varices, there is no standardized treatment for ectopic varices. For those, NBCA injection is also an option [7,14,15]. However, major complications, particularly systemic embolization, may occur when using such liquid agent.

The purpose of this study was to conduct a systematic review and meta-analysis of published studies to assess the safety, efficacy, and outcomes of NBCA-Lipiodol^®^ injection for the treatment of variceal GIB.

## 2. Materials and Methods

According to our country legislation, institutional review board approval was not required for this retrospective assessment of published data. The analysis was performed in compliance with the Preferred Reporting Items for Systematic Reviews and Meta-Analyses (PRISMA) guidelines [16]. 

### 2.1. Search Strategy

The MEDLINE/PubMed, EMBASE, and SCOPUS databases were searched to identify relevant studies published from January 1980 to December 2019. The search terms were: “(lipiodol OR oil OR ethiod OR poppyseed oil) AND (glue OR cyanoacrylate OR histoacryl OR nbca OR enbucrilate OR enbucrylate OR glubran) AND (bleeding OR hemorrhage) AND (gastrointestinal OR gi OR intestinal OR gastric) AND (embolization OR embolisation OR sclerotherapy OR embolization, therapeutic (MeSH Terms)) AND (human OR patient).” A manual search of reference lists of other studies and of articles from previous searches was performed, which led us to find a few additional studies. Duplicate publications were found by comparing author names, study dates, treatment comparisons, sample sizes, or outcomes and were then excluded.

### 2.2. Inclusion and Exclusion Criteria

The literature search and the selection of the eligible articles were conducted by 2 reviewers working independently of each other. Disagreements were resolved by consensus.

The inclusion criteria for the selected studies were as follows: (1) Original research article written in English; (2) human study subjects; (3) prospective and retrospective studies; (4) subjects underwent injection of NBCA-Lipiodol^®^ (Lipiodol^®^ Ultra Fluid, Guerbet, Aulnay-sous-Bois, France) mixture alone for GIB; (5) article showed outcomes of NBCA-Lipiodol^®^ mixture for at least 5 patients; (6) data and outcomes about both UGIB and LGIB were clearly identified and distinguishable. 

We excluded studies meeting the following criteria: (1) Case reports, abstracts, editorials, review articles, letters to the editor, chapters contained within a book, and preclinical studies; (2) articles reporting data on fewer than 5 patients; (3) articles reporting results from only combined embolic agents or results from different techniques or other techniques; (4) articles showing no clear results for NBCA-Lipiodol^®^ mixture injection in at least 5 patients or presenting duplicate results; (5) publications presenting data and results from UGIB and LGIB that were not clearly identified and distinguishable.

First, titles and abstracts of the articles were reviewed. Second, reviewers evaluated full-text articles for eligibility. Finally, we excluded articles that were related to nonvariceal GIB (data analyzed in a separate study).

### 2.3. Data Extraction and Definition

The following data were collected from the full-text articles that were included for analysis: First author, study country, publication year, study design (retrospective versus prospective, comparative or not, randomized or not), and bleeding site (UGIB or LGIB). For all included studies and each arm of comparative studies, we separately extracted the following data for both UGIB and LGIB: Number of analyzed patients, mean patient age, percentage of male patients, type of NBCA glue, NBCA-Lipiodol^®^ mixture ratio, technical success, 30-day rebleeding, and 30-day overall and major complications. For variceal bleeding, data regarding the varices type according to Sarin classification for gastroesophageal and isolated gastric varices [17] and the use of vasoactive drugs such as terlipressin, somatostatin, or octreotide, were also extracted.

The clinical endpoints were technical success, 30-day rebleeding, and 30-day overall and major complications as defined in the Society of Interventional Radiology guidelines [18]. Minor complications result in no consequence and no therapy or nominal therapy and include overnight admission for observation only. Major complications result in minor hospitalization (<48 h) and therapy, require major therapy and an unplanned increase in the level of care and prolonged hospitalization, or result in permanent adverse sequelae or death [18]. These endpoints were chosen due to strong heterogeneity among studies endpoints that did not allow the use of those recommended by the Baveno consensus [19].

### 2.4. Statistical Analyses

The technical success, 30-day rebleeding, 30-day overall complications, and 30-day major complications rates were reported for each study with their 95% confidence interval (95% CI) computed using the Clopper exact method. The estimated overall rates were computed with their 95% CI, based on each study rate weighted by the number of patients involved in each study, with random effect modeling. A forest plot was drawn for each rate of each study and for each overall estimated rate with their corresponding 95% CIs. The relative risks (i.e., rate ratio (RR)) of the technical success and 30-day rebleeding were evaluated for the individual randomized controlled trials (RCT) that compared NBCA-Lipiodol^®^ injection to another treatment method (comparator). Whenever possible, the RR was taken directly from the corresponding articles. Otherwise, the RR was calculated as follows:RR = (n_NBCA_Lipiodol_/N_NBCA_Lipiodol_)/(n_comparator_/N_comparator_)(1)
where: 

n is the number of patients with technical success (or 30-day rebleeding) in the corresponding treatment group and N is the total number of patients in the corresponding treatment group.

The RRs of the technical success rate and rebleeding for each RCT are presented in forest plots with the lower confidence limit (LCL) and upper confidence limit (UCL). 

The Q test and I^2^ statistic were used to evaluate heterogeneity across studies. A significant Q test indicated heterogeneity across studies. I² statistic values were interpreted as follows: 0% to 40%, heterogeneity might not be important; 30% to 60%, possible moderate heterogeneity; 50% to 90%, possible substantial heterogeneity; and 75% to 100%, considerable heterogeneity [20].

Forest plots were drawn for technical success and rebleeding, presenting the RR of each study and the overall RR estimate with their corresponding 95% CIs. 

Statistical analyses were performed using MedCalc for Windows, version 19.2.6 (MedCalc Software, Ostend, Belgium).

## 3. Results

### 3.1. Article Selection and Patient Characteristics

Figure 1 is the article flowchart. There were no duplicate studies. In total, 43 studies were included in the meta-analysis [10,11,21,22,23,24,25,26,27,28,29,30,31,32,33,34,35,36,37,38,39,40,41,42,43,44,45,46,47,48,49,50,51,52,53,54,55,56,57,58,59,60,61]. There were 15 prospective cohort studies [23,25,27,28,30,31,35,37,38,43,45,48,51,52,59], 8 single-arm cohort studies [23,30,31,35,43,45,51,52], 7 comparative studies [25,27,28,37,38,48,59], and 20 retrospective studies [10,21,22,24,26,32,33,34,36,42,44,46,47,49,50,53,55,56,60,61], including 4 retrospective comparative studies [10,46,50,53]. There were eight randomized controlled trials [11,29,39,40,41,54,57,58]. The study periods ranged from 1980–1996 to 2016–2017 [11,46]. A total of 3484 patients were included in these studies. Data about patients’ age and sex were not available in three [35,51,56] and one [35] studies, respectively. Among 3291 patients, the mean age was 54.9 years. Data about patients age were not available for 193 patients. Among 3449 patients, there were 2459 (71.3%) men and 990 (28.7%) women, respectively. Data about patients’ gender were missing for 35 patients. Table 1 reports the main characteristics of the studies and patients.

### 3.2. Types of NBCA Glue and NBCA-Lipiodol^®^ Ratio Used

NBCA was injected during endoscopy in 42 studies [10,11,21,22,23,24,25,26,27,28,29,30,31,32,33,34,35,36,37,38,39,40,41,42,43,44,45,46,47,48,49,50,51,52,53,54,55,56,57,58,59,61] and through a direct percutaneous approach for stomal varices in 1 study [60]. The types of NBCA glue used in the included studies were as follows: Histoacryl^®^ (B. Braun, Melsungen, Germany) was the most used NBCA glue (37 studies, 3282 patients, 94.2% of patients) [10,21,22,23,24,25,26,27,28,29,30,31,32,33,34,35,37,38,39,40,41,42,44,45,46,47,49,50,51,53,54,55,56,57,58,60,61]; and GluStitch^®^ Twist (GluStitch Inc., Delta, BC, Canada), LiquiBand^®^ (MedLogic Global Ltd., Plymouth, UK), and Glubran^®^2 (GEM Srl, Viareggio, Italy) were used, respectively, in only 1 (57 patients) [11], 1 (66 patients) [36], and 2 studies (27 patients) [52,59]. The type of NBCA glue used was not indicated in 2 studies, i.e., in 52 patients [43,48]. The most used NBCA-Lipiodol^®^ mixture ratio was 1:1 in 25 studies involving 2208 patients [10,21,22,23,24,27,30,31,32,33,35,36,38,41,43,48,50,51,52,53,57,58,59,61]. Other ratios were 1:1.4 in 4 studies (266 patients) [25,37,45,54], 1:1.6 in 7 studies (391 patients) [11,26,28,29,39,49,56], 1:1.8 in 2 studies (54 patients) [29,60], 1:1 or 1:1.6 in 1 study (21 patients) [47], 1:3 in 1 study (37 patients) [40], 2:1 in 1 study (97 patients) [34], and 2.3:1 in 1 study (17 patients) [46]. In 1 study, the ratio range was 1:1 to 1:1.5 (97 patients) [44]. The NBCA was diluted to a final concentration of 70% or 83% in 5% Lipiodol^®^ (228 patients) [55]. The NBCA-Lipiodol^®^ ratio was not mentioned in 1 study (68 patients) [42].

### 3.3. Technical Success and 30-Day Rebleeding Outcomes

In total, 37 studies reported data on technical success outcome [11,21,22,25,26,27,28,29,30,31,33,34,35,36,37,38,39,40,42,43,44,45,46,47,48,49,51,52,53,54,55,56,57,58,59,60,61]. Technical success was achieved in 2223 (94.1%) of 2341 patients (95% CI, 91.6–96.1%; Figure 2), with possible substantial to considerable heterogeneity across studies (*p* < 0.0001, I^2^ = 78.63%). 

Of the 43 studies, 37 reported data on the 30-day rebleeding rate. In total, 30-day rebleeding occurred in 599 (24.2%) of 3011 patients (95% CI, 18.9–29.9%) [11,21,22,23,24,25,27,28,29,30,31,32,33,34,35,36,37,38,39,40,43,44,45,47,48,49,50,51,52,53,54,56,57,58,59,60,61], with considerable heterogeneity across studies (*p* < 0.0001, I^2^ = 91.01%; Figure 3). Table 2 recapitulates the pooled technical success and 30-day rebleeding.

Characteristics of the RCT that compared NBCA-Lipiodol^®^ injection to another treatment method are summarized in Table 3. Data on the technical success rate were available in seven RCTs that compared NBCA-Lipiodol^®^ injection to another treatment method [11,39,40,41,54,57,58]. The average RR of the technical success rate was 1.13 (LCL, 0.99; UCL, 1.30). The RRs of the technical success rate for each RCT are presented in Figure 4a. Six RCT that compared NBCA-Lipiodol^®^ injection to another treatment method reported data on the 30-day rebleeding rate [11,39,40,54,57,58]. The average RR of the 30-day rebleeding rate was 0.83 (LCL, 0.61; UCL, 1.13). The RRs of the 30-day rebleeding rate for each RCT are presented in Figure 4b.

### 3.4. 30-Day Overall and Major Complication Rates

In total, 30 studies reported data on 30-day complications, including 3068 patients [10,11,22,23,24,25,26,27,28,29,30,31,33,34,35,36,37,38,40,41,43,44,45,46,47,48,49,50,51,52,53,56,58,59,60,61]. One-month overall complications occurred in 475 (15.9%) of 3068 patients (95% CI, 11.2–21.3%), with considerable heterogeneity across studies (*p* < 0.0001, I^2^ = 92.48%; Figure 5).

In total, 31 studies reported data on 1-month major complications, including 2634 patients [10,21,22,23,24,26,27,28,29,30,32,34,35,36,38,42,43,44,45,46,47,48,49,50,53,55,56,58,59,60,61]. One-month major complications occurred in 150 (5.3%) of 2634 patients (95% CI, 3.3–7.8%), with possible substantial to considerable heterogeneity across studies (*p* < 0.0001, I^2^ = 82.52%; Figure 6). Table 2 recapitulates the complications rates for each study.

## 4. Discussion

The present meta-analysis of 43 studies, which involves 3484 patients, demonstrates that the use of NBCA-Lipiodol^®^ mixture is safe and efficient for variceal GIB patients. A very high technical success rate of 94.1% patients (95% CI, 91.6–96.1%), a moderate 30-day rebleeding rate of 24.2% (95% CI, 18.9–29.9%), and a low risk of 30-day major complications of 5.3% (95% CI, 3.3–7.8%) were found in our study. NBCA-Lipiodol^®^ mixture was injected during endoscopy in all studies but one in which it was injected through a direct percutaneous approach for stomal varices [60]. In addition, the average RR of the technical success and 30-day rebleeding rates of RCT included that compared NBCA-Lipiodol^®^ injection to another treatment method favored NBCA-Lipiodol^®^ injection, with averages RR of 1.13 (LCL, 0.99; UCL, 1.30) and 0.83 (LCL, 0.61; UCL, 1.13), respectively.

EASL recommends EVL once variceal bleeding is confirmed by endoscopy and sclerotherapy when ligation is not feasible [8]. For esophageal variceal hemorrhage, EVL is recommended by the AASLD [7]. This is supported by a recent meta-analysis that showed that EVL is superior to sclerotherapy in this setting, with EVL being associated with a significant improvement in bleeding control when compared to sclerotherapy (RR = 1.08; 95 % CI, 1.02–1.15) [62]. However, only one trial comparing EVL to cyanoacrylate injection was included in this article and showed no difference in terms of efficacy, rebleeding rate, or mortality [39]. For gastric varices, EVL should only be performed for small varices in which the complete vessel can be suctioned into the ligation device [8]. A meta-analysis suggested that endoscopic cyanoacrylate injection and EVL are equally effective for initial hemostasis of bleeding gastric varices, while cyanoacrylate may be more efficient for preventing rebleeding [13]. However, the quality of the evidence remained very low [13].

For all outcomes, our analysis showed a significant heterogeneity in the results across studies. It could be partially explained by the variability regarding the bleeding site, varix types, and NBCA-Lipiodol^®^ ratio. The endoscopic management of variceal bleeding depends on the type of varices concerned by the hemorrhage. For esophageal and GOV1 varices, sclerotherapy is classically considered as a second-choice treatment when EVL is not feasible [7,8]. For GOV2 and IGV1 varices, sclerotherapy is more appropriate as first-line treatment. However, most of the included studies did not report sub-group results according to the type of varices. In addition, variability in patient characteristics, particularly cirrhosis and portal hypertension stages, could have impacted the results, especially the rebleeding rates. 

In cases of endoscopic and/or pharmacological treatment failure with persistent uncontrollable bleeding, TIPS can be used as a rescue treatment by allowing a significant decrease or even normalization of the portal pressure and has demonstrated good results for bleeding control [3,5,7,63]. However, the prognosis depends on the general condition, the liver function reserve, and the associated comorbidities of the patient [64,65,66,67]. Current evidence supports the early use of TIPS for patient with cirrhosis and acute variceal bleeding [68,69]. Transjugular embolization of the varices at the time of TIPS can also be performed [3,4,6,70,71,72]. This approach might reduce the risk of variceal rebleeding for patients with gastroesophageal varices [70,71,73,74]. In addition, a trial demonstrated that the 6-month shunt patency was significantly higher (96.2% vs. 82%, *p* = 0.019) when TIPS was combined with varices embolization [74]. Furthermore, a study found that persistence of esophageal or gastric varices on trans-TIPS angiographic control was associated with increased shunt revision rates of 13%, 26.3%, and 36.3% at 3, 12, and 24 months, respectively [75]. The choice of the best embolic agent, though, is still under debate. Different agents, such as vascular coils, vascular occlude, and liquid embolic agents such as NBCA or ethylene vinyl alcohol-based copolymers (Onyx^®^ or Squid^®^), can be used [3,4,6,70,71,72,76]. Lakhoo et al. demonstrated that most gastric varices showed persistent patency despite TIPS decompression and variceal embolization using mechanical agents, metallic coils, and/or plugs (61% with varices embolization at a median of 128.5 days after TIPS creation) [77]. In contrast, Shi et al. compared the use of TIPS alone versus TIPS with adjunctive embolotherapy using cyanoacrylate regarding recurrent hemorrhage following TIPS insertion [71]. The probability of absence of rebleeding at 1, 3, and 5 years and the probability of hepatic encephalopathy were significantly lower in the TIPS + embolization group (*p* = 0.042, 0.048, and 0.019, respectively) as compared to the TIPS alone group [72]. Therefore, the use of liquid agents, such as NBCA, could improve the outcomes. 

There is no consensus concerning the best therapy of bleeding ectopic varices due to heterogeneous localization and anatomy [7]. Therefore, patients should be evaluated and treated on a case-by-case basis. Local treatments of ectopic varices can be difficult or even impossible. TIPS procedure might represent a good approach in this setting [4,5,6]. Some authors have suggested that transcatheter varices embolization using NBCA with or without TIPS placement could be a useful option for bleeding ectopic varices [78]. In the case of stomal varix bleeding, direct percutaneous approach with NBCA injection demonstrated good results [60].

BRTO is currently recognized as an alternative to TIPS for treatment of fundal varices associated with a large gastro/splenorenal collateral when the patient is not an appropriate candidate for TIPS because of hepatic encephalopathy or poor hepatic reserve [15,79]. A recent trial showed that BRTO was more effective than endoscopic cyanoacrylate injection in preventing rebleeding from gastric variceal bleeding [80]. During the BRTO procedure, sclerosing agents, such as ethanolamine oleate or sodium tetradecyl sulfate mixed with water-soluble contrast media or Lipiodol^®^, are used for gastric varix obturation [81,82]. Foam sclerosant of Sotradecol mixed with gas and ethiodized oil has also been used [83]. Small collateral veins are generally embolized prior to sclerosing agent injection to prevent leakage of the sclerosing agent or varix recurrence [81,82,83,84]. Tsuruya et al. reported a case where NBCA-Lipiodol^®^ were used for gastrorenal shunt embolization after injecting sclerosing agent in a severely obese patient, resulting in a shorter procedure time [85]. 

Varices embolization through transsplenic route has also been reported [86]. Percutaneous transhepatic or transjugular intrahepatic access to the portal vein is not always feasible or can be difficult, for instance, in settings of portal vein occlusion, portal vein compression by perihepatic extensive hematoma, attenuated intrahepatic portal vein, or cavernous transformation of the portal vein. BRTO is also not always feasible, since this method requires the presence of a gastrorenal shunt. Percutaneous transsplenic approach is another way to access the portal venous system and can be useful in these specific cases. In the study by Chu et al., gastric or jejunal varices embolization through transsplenic route using NBCA-Lipiodol^®^ mixture was performed successfully in all four patients who presented hematemesis or hematochezia, with no observed bleeding recurrence during the follow-up period [86].

NBCA-Lipiodol^®^ has several advantages. Since the NBCA-Lipiodol^®^ ratio impacts the liquid viscosity and the time of polymerization, the operator can adjust it to the blood flux, allowing a distal embolization. Lipiodol^®^ makes the mixture radio-opaque, allowing easier control when injected under fluoroscopic control. Its liquid nature allows diffusion through collateral vessels, which might lead to less recurrence. In addition to mechanical obstruction and thrombosis, NBCA acts as a sclerosing agent, inducing chemical phlebitis, fibrosis, and complete destruction of the vein. In another setting, NCBA has demonstrated lower recurrence rates than mechanical agents (coils and/or plugs) and sclerosing agent (polidocanol) for transcatheter retrograde varicocele embolization, with a shorter procedure time [87]. In addition, the polymerization of NBCA in contact with blood is independent of the coagulation status of the patient and may therefore be more efficient than other embolic agents in patients who present coagulopathy [88].

The strengths of this study include the comprehensive literature search strategy and robust methodology, as well as the reporting of results in compliance with PRISMA guidelines. However, our analysis had several limitations. First, among the 43 included studies, 20 were retrospective case series. Second, heterogeneity in the bleeding site and varix types occurred across studies. It is well known that the choice of the best therapy may depend on the bleeding site and varix type. Unfortunately, subgroup analyses were not reported in most included studies. Therefore, separate analysis according to the type of varices was not feasible due to these missing data. However, our study aimed to focus on the use of NBCA-Lipiodol^®^ for all variceal GIBs. Third, the chosen clinical endpoints in the present meta-analysis differed from those recommended by the Baveno consensus [19]. Considerable variability was found regarding the endpoints and their definitions among the included studies. This variability can be partially explained by the retrospective nature of most of the included studies, in which missing data could have led to the inability to use the recommended Baveno outcomes. Also, many of the included studies were published before the last Baveno consensus. In addition, Baveno consensus recommendations have evolved since the first meeting in 1990 (Baveno I). 

This point is very interesting, and our meta-analysis reflects the heterogeneity of real-life reported results on this topic despite recommendations, which represents the strength of this meta-analysis in our opinion. It is of utmost importance for readers to be aware of this discrepancy between real-life reported results with this technique and the recommendations on it, meaning that upcoming studies on this topic should be more rigorous with these criteria. Our analysis was indeed based on real-life reported data. We attempted to minimize the impact of this variability by applying similar definitions for the main outcomes of interest. Fourth, data regarding the stages of cirrhosis and portal hypertension were not collected and analyzed, which could have impacted the results. Fifth, variations also occurred in the type of NBCA glue and in the NBCA-Lipiodol^®^ mixture ratio. Sixth, significant heterogeneity for all outcomes was found in the results across studies. However, a random effect model was used to minimize this source of bias. Seventh, publication bias might have occurred, as negative results are commonly not published. Last, the quality of the included studies was not evaluated.

## 5. Conclusions

The present meta-analysis of 43 studies involving 3484 patients demonstrates that the use of NBCA-Lipiodol^®^ mixture for variceal GIB patients is safe and effective, with a very high technical success rate, moderate rebleeding rate, and low risk of major complications.

## Figures and Tables

**Figure 1 jcm-10-02298-f001:**
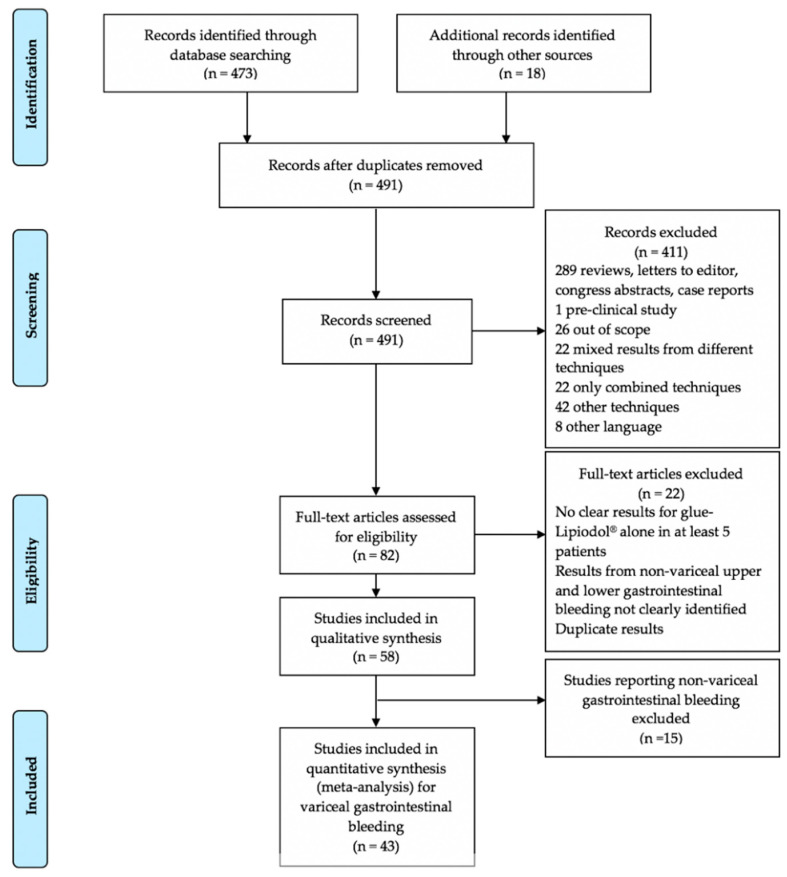
Preferred Reporting Items for Systematic Reviews and Meta-Analyses (PRISMA) flow diagram of the article selection process.

**Figure 2 jcm-10-02298-f002:**
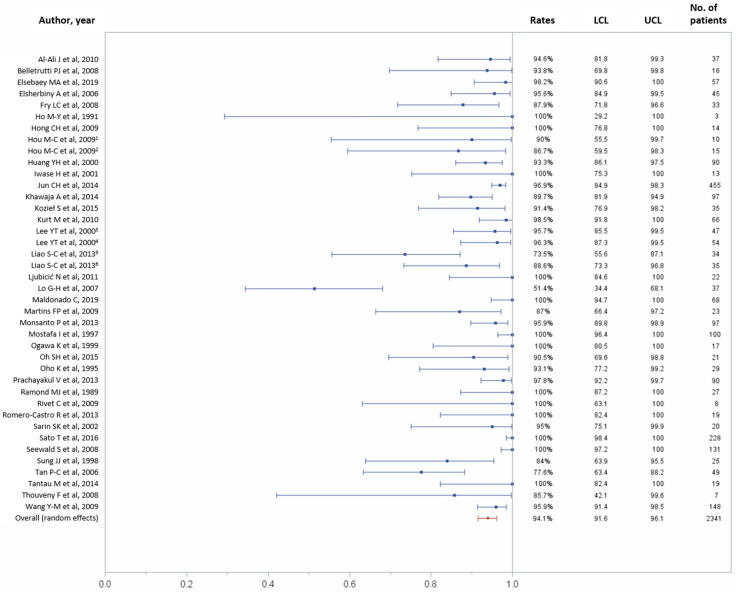
Forest plot of the technical success rate. Overall rate (random effects model): 94.1% (95% CI, 91.6–96.1%); heterogeneity: Q = 182.49, *p* < 0.0001, I² = 78.63%; UCL, upper control limit; LCL, lower control limit; No, number.

**Figure 3 jcm-10-02298-f003:**
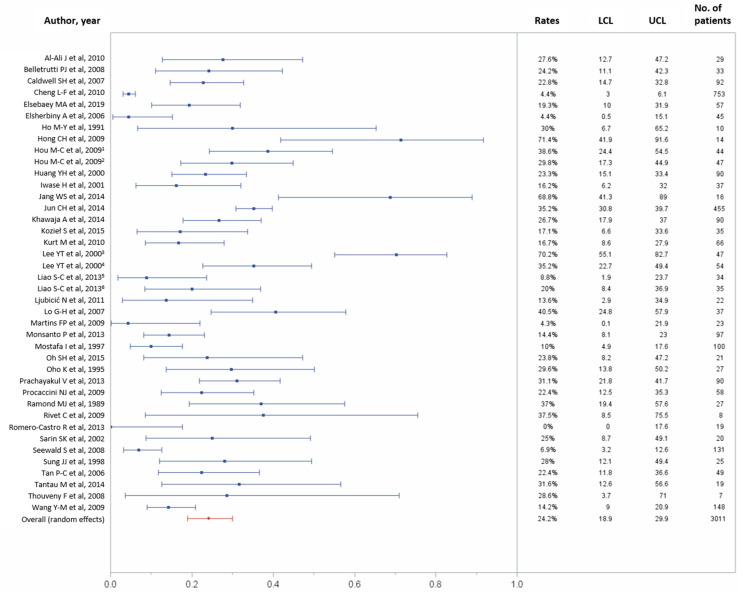
Forest plot of the 30-day rebleeding rate. Overall rate (random effects model): 24.2% (95% CI, 18.9–29.9%); heterogeneity: Q = 434.05, *p* < 0.0001, I² = 91.01%; UCL, upper control limit; LCL, lower control limit; No, number.

**Figure 4 jcm-10-02298-f004:**
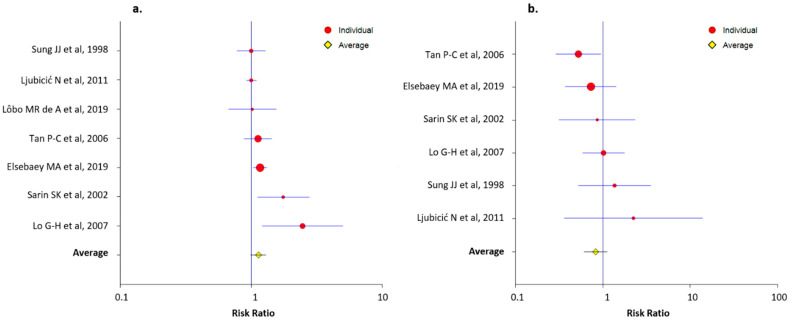
Forest plots presenting the risk ratios of the technical success (**a**) and the 30-day rebleeding (**b**) rates for each randomized controlled trial that compared NBCA-Lipiodol^®^ injection to another treatment method.

**Figure 5 jcm-10-02298-f005:**
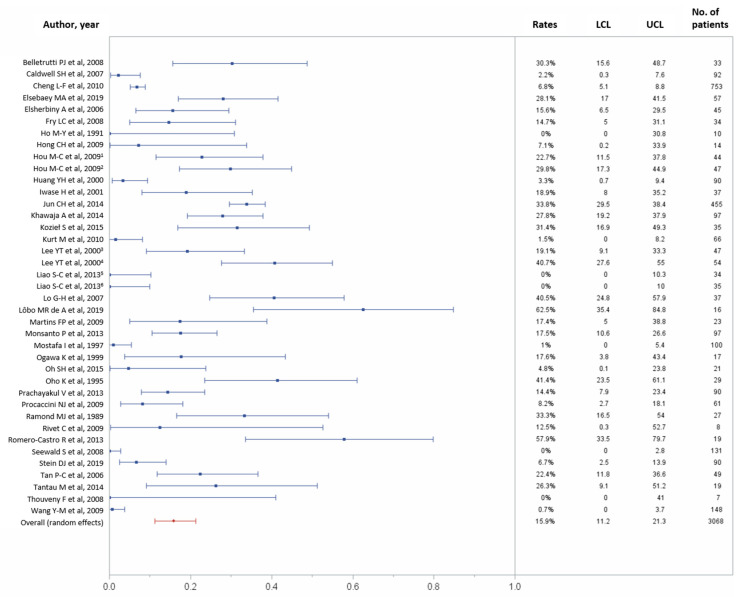
Forest plots of the 30-day overall complications rates. Overall rate (random effects model): 15.9% (95% CI, 11.2–21.3%); heterogeneity: Q = 505.22, *p* < 0.0001, I^2^ = 92.48%; UCL, upper control limit; LCL, lower control limit; No, number.

**Figure 6 jcm-10-02298-f006:**
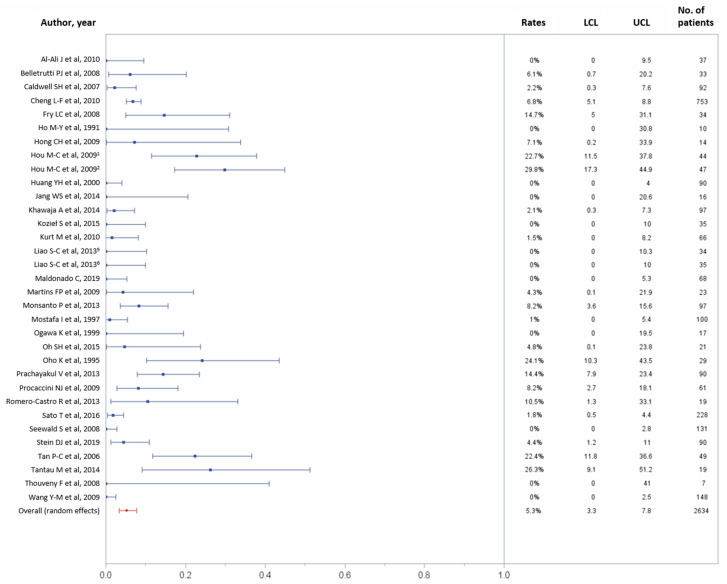
Forest plots of the 30-day major complications rates. Overall rate (random effects model): 5.3% (95% CI, 3.3–7.8%); heterogeneity: Q = 183.02, *p* < 0.0001, I^2^ = 82.52%; UCL, upper control limit; LCL, lower control limit; No, number.

**Table 1 jcm-10-02298-t001:** Characteristics of the studies included in the meta-analysis.

	Study Type	Study Period	Country	No. of Patients *	Gender * (Male, %)	Mean Age * (Years)	GIB Site	Sarin Classification of GV (%)	Type of Glue	NBCA-Lipiodol^®^ Ratio	Use of Vasoactive Drugs **
Al-Ali J et al., 2010 [21]	R	2001–2006	Canada	37	59.5	60	GV	NS	Histoacryl^®^	1:1	For 1 patient
Belletrutti PJ et al., 2008 [22]	R	2001–2006	Canada	34	55.9	54.5	GV	GOV1 (70.2), GOV2 (6.4), IGV1 (23.4)	Histoacryl^®^	1:1	NS
Caldwell SH et al., 2007 [23]	P	NS	USA	92	71.7	55	GV	GOV1 (14.1), GOV2 (53.3), IGV1 (29.3), IGV2 (3.3)	Histoacryl^®^	1:1	Yes
Cheng LF et al., 2010 [24]	R	1996–2006, 2003–2007	China	753	72.6	51	GV	GOV1 (36.4), GOV2 (25.2), GOV1–2 (19.4), IGV1 (19.0)	Histoacryl^®^	1:1	NS
Elsebaey MA et al., 2019 [11]	RCT	2016–2017	Egypt	57	68.4	55	EV		NBCA (GluStitch^®^)	1:1.6	Yes
Elsherbiny A et al., 2006 [25]	PC	2004	Egypt	45	64.4	57.6	GV	GOV1 (NS) GOV2 (NS)	Histoacryl^®^	1:1.4	NS
Fry LC et al., 2008 [26]	R	5-years period	Germany	33	48.5	54	GV	GOV1 (20), GOV2 (80)	Histoacryl^®^	1:1.6	Yes
Ho MY et al., 1991 [27]	PC	1990–1991	China	10	80.0	60	GV and combined GV + EV	NS	Histoacryl^®^	1:1	NS
Hong CH et al., 2009 [28]	PC	2005–2007	Korea	14	85.7	55.4	GV	GOV2 (64.3), GOV1 + GOV 2 (14.3), IGV1 (21.4)	Histoacryl^®^	1:1.6	Yes
Hou MC et al., 2009 [29] ^1^	RCT	2005–2007	Taiwan	47	72.3	59.19	GV	GOV1 (46.8), GOV2 (34.0), IGV1 (19.2)	Histoacryl^®^	1:1.8	Yes
Hou MC et al., 2009 [29] ^2^				44	86.4	59.89	GV	GOV1 (36.4), GOV2 (27.3), IGV1 (36.4)	Histoacryl^®^	1:1.6	Yes
Huang YH et al., 2000 [30]	P	1992–1998	Taiwan	90	77.8	58	GV	NS	Histoacryl^®^	1:1	NS
Iwase H et al., 2001 [31]	P	1992–1999	USA and Japan	37	64.9	61	GV	IGV1 (100)	Histoacryl^®^	1:1	No
Jang WS et al., 2014 [32]	R	2008–2012	Korea	16	56.3	61.8	GV	NS	Histoacryl^®^	1:1	NS
Jun CH et al., 2014 [33]	R	2004–2013	Korea	455	83.3	57.65	GV	GOV1 (61.5), GOV2 (38.5)	Histoacryl^®^	1:1	NS
Khawaja A et al., 2014 [34]	R	1998–2011	Pakistan	97	63.9	51	GV	GOV1 (20.6), GOV2 (36.1), IGV1 (41.2), IGV2 (2.1)	Histoacryl^®^	2:1	Yes
Kozieł S et al., 2015 [35]	P	2013–2015	Poland	35	NS	NS	GV	GOV2 (28.6), IGV1 (68.6), IGV2 (2.9)	Histoacryl^®^	1:1	Ns
Kurt M et al., 2010 [36]	R	2004–2010	Turkey	66	62.1	52	GV	GOV1 (4.5), GOV2 (69.7), IGV1 (25.8)	Liquiband^®^	1:1	Yes
Lee YT et al., 2000 [37] ^3^	PC	1993–1998	China	54	63.0	61	GV	GOV1 (37.0), GOV2 (33.3), IGV1 (29.6)	Histoacryl^®^	1:1.4	Yes
Lee YT et al., 2000 [37] ^4^				47	74.5	59	GV	GOV1 (34.0), GOV2 (42.6), IGV1 (23.4)	Histoacryl^®^	1:1.4	Yes
Liao SC et al., 2013 [38] ^5^	PC	2001–2007	Taiwan	35	45.7	61	GV	NS	Histoacryl^®^	1:1	Yes
Liao SC et al., 2013 [38] ^6^				34	52.9	59	GV	NS	Histoacryl^®^	1:1	Yes
Ljubicić N et al., 2011 [39]	RCT	2004–2008	Croatia	22	72.7	57	EV		Histoacryl^®^	1:1.6	Yes
Lo GH et al., 2007 [40]	RCT	1999–2004	Taiwan	37	75.7	52	GV	GOV1 (45.9), GOV2 (51.4), IGV (2.7)	Histoacryl^®^	1:3	Yes
Lôbo MR de A et al., 2019 [41]	RCT	2014–2016	Brazil	16	31.3	57.7	GV	GOV2 (81.2), IGV1 (18.8)	Histoacryl^®^	1:1	NS
Maldonado C, 2019 [42]	R	2011–2017	Colombia	68	50.0	64	FV		Histoacryl^®^	NS	NS
Martins FP et al., 2009 [43]	P	NS	Brazil	23	65.2	53.4	GV	GOV2 (87.0), IGV1 (13.0)	NBCA	1:1	NS
Monsanto P et al., 2013 [44]	R	1998–2010	Portugal	97	80.4	59.6	GV	GOV1 (37.2), GOV2 (27.8), IGV1 (30.9), IGV2 (4.1)	Histoacryl^®^	1:1 to 1:1.5	Yes
Mostafa I et al., 1997 [45]	P	NS	Egypt	100	84.0	44.7	GV	GOV1 (20), GOV2 or IGV1 (80)	Histoacryl^®^	1:1.4	NS
Ogawa K et al., 1999 [46]	RC	1980–1996	Japan	17	76.5	63.5	GV	NS	Histoacryl^®^	2.3:1	NS
Oh SH et al., 2015 [47]	R	2004–2011	South Korea	21	38.1	8.7	GV	GOV1 (76.2), GOV2 (23.8)	Histoacryl^®^	1:1 or 1:1.6	NS
Oho K et al., 1995 [48]	PC	1989–1992	Japan	29	72.4	57	CV (*n* = 12), FV (*n* = 17)		NBCA	1:1	No
Prachayakul V et al., 2013 [49]	R	2008–2011	Thailand	90	74.4	55.9	GV	GOV1 (44.4), GOV2 (33.3), IGV1 (21.2), IGV2 (1.1)	Histoacryl^®^	1:1.6	Most patients
Procaccini NJ et al., 2009 [50]	RC	1997–2007	USA	61	70.5	54.5	GV	NS	Histoacryl^®^	1:1	Majority of patients
Ramond MJ et al., 1989 [51]	P	1984–1988	France	27	63.0	NS	GV	NS	Histoacryl^®^	1:1	NS
Rivet C et al., 2009 [52]	P	2001–2005	France	8	25.0	1.3	GV	GOV (87.5), IGV (12.5)	Glubran2^®^	1:1	NS
Romero-Castro R et al., 2013 [53]	RC	2008–2012	Spain and Germany	19	73.7	60.8	GV	GOV1 (5.3), GOV2 (47.4), IGV1 (47.4)	Histoacryl^®^	1:1	NS
Sarin SK et al., 2002 [54]	RCT	1995–1998	India	20	75.0	36.1	GV	IGV (100)	Histoacryl^®^	1:1.4	Yes
Sato T et al., 2016 [55]	R	NS	Japan	228	64.5	62.5	GV (*n* = 221)	GOV2 (48.9), IGV1 (47.1), IGV2 (4.1)	Histoacryl^®^	#	Not routinely used
							DV (*n* = 5)				
							AV (*n* = 2)				
Seewald S et al., 2008 [56]	R	1994–2003	Germany and Egypt	131	69.5	NS	GV	GOV2 (17.6), IGV1 (82.4)	Histoacryl^®^	1:1.6	NS
Stein DJ et al., 2019 [10]	RC	1997–2015	USA	90	70.0	55.9	GV	GOV2 (25.6), IGV1 (74.4)	Histoacryl^®^	1:1	NS
Sung JJ et al., 1998 [57]	RCT	NS	Hong Kong	25	88.0	49.8	EV		Histoacryl^®^	1:1	Yes
Tan PC et al., 2006 [5]	RCT	1996–2002	Taiwan	49	71.4	61.35	GV	GOV1 (55.1), GOV2 (18.4), IGV1 (26.5)	Histoacryl^®^	1:1	Yes
Tantau M et al., 2014 [59]	PC	2010–2012	Romania	19	52.6	62.3	GV	GOV1 (57.9), GOV2 (42.1)	Glubran^®^	1:1	Yes
Thouveny F et al., 2008 [60]	R	1998–2006	France	7	71.4	69	SV		Histoacryl^®^	1:8	NS
Wang YM et al., 2009 [61]	R	2007–2008	China	148	73.0	50.1	U	GOV1 (45.9), GOV2 (33.1), IGV1 (20.3), IGV2 (1)	Histoacryl^®^	1:1	Yes

^1^ 1.0 mL NBCA; ^2^ 0.5 mL NBCA; ^3^ Biweekly endosonography followed by repeated injections of cyanoacrylate; ^4^ On demand cyanoacrylate injections; ^5^ Traditional endoscopic follow-up (control group); ^6^ Miniature ultrasound probe sonography; * patients treated with NBCA-Lipiodol^®^ mixture; ** vasoactive drugs such as somatostatin, terlipressin or octreotide; # NBCA diluted to a final concentration of 70% or 83% in 5% Lipiodol^®^; No., number; RCT, randomized control trial; R, retrospective; P, prospective; PC, prospective comparative, RC, retrospective comparative; NS, not specified; GIB, gastrointestinal bleeding; EV, esophageal varices; GV, gastric varices; CV, cardiac varices; FV, fundic varices; DV, duodenal varices; AV, anastomotic varices after choledochojejunostomy; SV, stromal varices; GOV, gastroesophageal varices; IGV, isolated gastric varices; NBCA, N-butyl cyanoacrylate. Histoacryl^®^ (B. Braun, Melsungen, Germany); GluStitch^®^ Twist (GluStitch Inc., Delta, BC, Canada); LiquiBand^®^ (MedLogic Global Ltd., Plymouth, UK); Glubran^®^2 (GEM Srl, Viareggio, Italy).

**Table 2 jcm-10-02298-t002:** Technical success, 30-day rebleeding and complications rates by study in patients treated with NBCA-Lipiodol^®^ mixture for variceal gastrointestinal bleeding.

Author, Year	Technical Success			30-Day Rebleeding			Complications				
	No. of Patients Evaluated	No. of Patients with Technical Success	Technical Success Rate (%)	No. of Patients Evaluated	No. of Patients with 30-Day Rebleeding	30-Day Rebleeding Rate (%)	No. of Patients Evaluated	1-Month Overall Complications		1-Month Major Complications	
								No. of Patients	Rate (%)	No. of Patients	Rate (%)
Al-Ali J et al., 2010 [21]	37	35	94.6	29	8	27.6	37			0	0.0
Belletrutti PJ et al., 2008 [22]	16	15	93.8	33	8	24.2	33	10	30.3	2	6.1
Caldwell SH et al., 2007 [23]				92	21	22.8	92	2	2.2	2	2.2
Cheng LF et al., 2010 [24]				753	33	4.4	753	51	6.8	51	6.8
Elsebaey MA et al., 2019 [11]	57	56	98.2	57	11	19.3	57	16	28.1		
Elsherbiny A et al., 2006 [25]	45	43	95.6	45	2	4.4	45	7	15.6		
Fry LC et al., 2008 [26]	33	29	87.9				34	5	14.7	5	14.7
Ho MY et al., 1991 [27]	3	3	100	10	3	30.0	10	0	0.0	0	0.0
Hong CH et al., 2009 [28]	14	14	100	14	10	71.4	14	1	7.1	1	7.1
Hou MC et al., 2009 [29] ^1^	10	9	90.0	44	17	38.6	44	10	22.7	10	22.7
Hou MC et al., 2009 [29] ^2^	15	13	86.7	47	14	29.8	47	14	29.8	14	29.8
Huang YH et al., 2000 [30]	90	84	93.3	90	21	23.3	90	3	3.3	0	0.0
Iwase H et al., 2001 [31]	13	13	100	37	6	16.2	37	7	18.9		
Jang WS et al., 2014 [32]				16	11	68.8	16			0	0.0
Jun CH et al., 2014 [33]	455	441	96.9	455	160	35.2	455	154	33.8		
Khawaja A et al., 2014 [34]	97	87	89.7	90	24	26.7	97	27	27.8	2	2.1
Kozieł S et al., 2015 [35]	35	32	91.4	35	6	17.1	35	11	31.4	0	0.0
Kurt M et al., 2010 [36]	66	65	98.5	66	11	16.7	66	1	1.5	1	1.5
Lee YT et al., 2000 [37] ^3^	54	52	95.7	54	19	70.2	54	22	19.1		
Lee YT et al., 2000 [37] ^4^	47	45	96.3	47	33	35.2	47	9	40.7		
Liao SC et al., 2013 [38] ^5^	34	25	73.5	34	3	8.8	34	0	0.0	0	0.0
Liao SC et al., 2013 [38] ^6^	35	31	88.6	35	7	20.0	35	0	0.0	0	0.0
Ljubicić N et al., 2011 [39]	22	22	100.0	22	3	13.6	-				
Lo GH et al., 2007 [40]	37	19	51.4	37	15	40.5	37	15	40.5		
Lôbo MR de A et al., 2019 [41]							16	10	62.5		
Maldonado C, 2019 [42]	68	68	100				68			0	0.0
Martins FP et al., 2009 [43]	23	20	87.0	23	1	4.3	23	4	17.4	1	4.3
Monsanto P et al., 2013 [44]	97	93	95.9	97	14	14.4	97	17	17.5	8	8.2
Mostafa I et al., 1997 [45]	100	100	100	100	10	10.0	100	1	1.0	1	1.0
Ogawa K et al., 1999 [46]	17	17	100				17	3	17.6	0	0.0
Oh SH et al., 2015 [47]	21	19	90.5	21	5	23.8	21	1	4.8	1	4.8
Oho K et al., 1995 [48]	29	27	93.1	27	8	29.6	29	12	41.4	7	24.1
Prachayakul V et al., 2013 [49]	90	88	97.8	90	28	31.1	90	13	14.4	13	14.4
Procaccini NJ et al., 2009 [50]				58	13	22.4	61	5	8.2	5	8.2
Ramond MJ et al., 1989 [51]	27	27	100	27	10	37.0	27	9	33.3		
Rivet C et al., 2009 [52]	8	8	100	8	3	37.5	8	1	12.5		
Romero-Castro R et al., 2013 [53]	19	19	100	19	0	0.0	19	11	57.9	2	10.5
Sarin SK et al., 2002 [54]	20	19	95.0	20	5	25.0					
Sato T et al., 2016 [55]	228	228	100				228			4	1.8
Seewald S et al., 2008 [56]	131	131	100	131	9	6.9	131	0	0.0	0	0.0
Stein DJ et al., 2019 [10]							90	6	6.7	4	4.4
Sung JJ et al., 1998 [57]	25	21	84.0	25	7	28.0					
Tan PC et al., 2006 [58]	49	38	77.6	49	11	22.4	49	11	22.4	11	22.4
Tantau M et al., 2014 [59]	19	19	100	19	6	31.6	19	5	26.3	5	26.3
Thouveny F et al., 2008 [60]	7	6	85.7	7	2	28.6	7	0	0.0	0	0.0
Wang YM et al., 2009 [61]	148	142	95.9	148	21	14.2	148	1	0.7	0	0.0

^1^ 1.0 mL NBCA; ^2^ 0.5 mL NBCA; ^3^ Biweekly endosonography followed by repeated injections of cyanoacrylate; ^4^ On demand cyanoacrylate injections; ^5^ Traditional endoscopic follow-up (control group); ^6^ Miniature ultrasound probe sonography. No., number; GIB, gastrointestinal bleeding; NBCA, N-butyl cyanoacrylate.

**Table 3 jcm-10-02298-t003:** Characteristics of the randomized controlled trials included in the meta-analysis that compared NBCA-Lipiodol^®^ injection to another treatment method (comparator).

Author, Year	Bleeding Site	NBCA-Lipiodol^®^	Comparator	
		No. of Patients	Treatment Method	No. of Patients
Ljubicić N et al., 2011 [39]	EV	22	EVL	21
Lo GH et al., 2007 [40]	GV	37	TIPS	35
Lôbo MR de A et al., 2019 [41]	GV	16	EUS-guided coils plus cyanoacrylate-Lipiodol^®^ mixture	16
Sarin SK et al., 2002 [54]	GV	20	Sclerotherapy with alcohol	17
Sung JJ et al., 1998 [57]	EV	25	Sclerotherapy with sodium tetradecyl sulphate	25
Tan PC et al., 2006 [58]	GV	49	EVL	48
Elsebaey MA et al., 2019 [11]	EV	57	Sclerotherapy with ethanolamine oleate	56

No, number; EV, esophageal varices; GV, gastric varices; EVL, endoscopic variceal ligation; TIPS, transjugular intrahepatic portosystemic shunt; EUS, endoscopic ultrasound.

## Data Availability

Data are contained within the article.

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
