# Peer review of "Safety, Efficacy, and Outcomes of N-Butyl Cyanoacrylate Glue Injection through the Endoscopic or Radiologic Route for Variceal Gastrointestinal Bleeding: A Systematic Review and Meta-Analysis"

_jcm, 2021, doi:10.3390/jcm10112298_

Round 1

Reviewer 1 Report

The authors provide a comprehensive review of the literature in the specific field of variceal embolization.  In my opinion it should be stated also in the title and abstract that embolization is performed in most cases (as emerges from the analisys) through the endoscopic route to orient the reader. The interventional radiology reader could be disoriented since the authors state only in the discussion section that: "NBCA-Lipiodol® mixture was injected during endoscopy in all studies but one in which it was injected through a direct percutaneous approach for stomal varices".

The authors should cite in the introduction the trans-splenic route to varices embolization which has been reported in the interventional radiology setting. Interestingly the paper "Percutaneous transsplenic access to the portal vein for management of vascular complication in patients with chronic liver disease; Hee Ho Chu DOI: 10.1007/s00270-011-0311-y" was not considered in the meta-analisys. Why? 

Moreover, in the discussion the authors describe the use of trans-TIPS glue embolization but again no paper about this technique was considered in the meta-analysis. Why?

As the authors state one of the main weakness of this work, apart from heterogeneity of methods and techniques, regards the variability of clinical situations which has not been investigated.

The manuscript is well written, it is clear and the statistical analisys seems appropriate. No major concerns with pubblication i n the present form.

Author Response

Response to Reviewer 1

  1. English language and style are fine/minor spell check required.

Reply: Thank you very much for your comment. English language has been checked by a native speaker as suggested.

  1. The authors provide a comprehensive review of the literature in the specific field of variceal embolization.  In my opinion it should be stated also in the title and abstract that embolization is performed in most cases (as emerges from the analysis) through the endoscopic route to orient the reader. The interventional radiology reader could be disoriented since the authors state only in the discussion section that: "NBCA-Lipiodol® mixture was injected during endoscopy in all studies but one in which it was injected through a direct percutaneous approach for stomal varices".

Reply: Thank you very much for your comment. We fully agree. It has been stated in the title and abstract that embolization is performed in most cases through the endoscopic route to orient the reader, as suggested.

  1. The authors should cite in the introduction the trans-splenic route to varices embolization which has been reported in the interventional radiology setting. Interestingly the paper "Percutaneous transsplenic access to the portal vein for management of vascular complication in patients with chronic liver disease; Hee Ho ChuDOI: 1007/s00270-011-0311-y" was not considered in the meta-analysis. Why? 

Reply: Thank you very much for your comment. This article is indeed very interesting. It was reviewed but excluded from the final analysis. In this study, only 4 patients underwent glue-lipiodol embolization for variceal bleeding. Due to the inclusion criteria [“5) article presented outcomes of NBCA-Lipiodol® mixture for at least 5 patients”] and exclusion criteria [“2) publications that reported data on fewer than 5 patients”] of the presented meta-analysis, this study was excluded. On the other hand, this method has been cited in the introduction as suggested (Lines 56-57): “Acute variceal GIB can be managed through various methods used alone or in combination: endoscopic therapy, the use of vasoactive drugs, balloon tamponade, endoscopically self-expandable metal stent placement, esophageal transaction, transjugular intra-hepatic portosystemic shunt (TIPS) with or without varices embolization, balloon-occluded retrograde transvenous obliteration (BRTO) and varices embolization through trans-splenic route.” The technique has also been discussed in a new paragraph in the discussion section (with the citation of this article; Lines 349-359), as suggested and as follows: “Varices embolization through transsplenic route has also been reported [83]. Percutaneous transhepatic or transjugular intrahepatic access to the portal vein is not always feasible or can be difficult, for instance in settings of portal vein occlusion, portal vein compression by perihepatic extensive hematoma, attenuated intrahepatic portal vein or cavernous transformation of the portal vein. BRTO is also not always feasible, since this method requires the presence of a gastrorenal shunt. Percutaneous transsplenic approach is another way to access the portal venous system and can be useful in these specific cases. In the study by Chu et al., gastric or jejunal varices embolization through trans-splenic route using NBCA-Lipiodol® mixture was performed successfully in all the four patients who presented hematemesis or hematochezia, with no observed bleeding recurrence during the follow-up period [85].”

  1. Moreover, in the discussion the authors describe the use of trans-TIPS glue embolization but again no paper about this technique was considered in the meta-analysis. Why?

Reply: Thank you very much for your comment. Unfortunately, studies that reported trans-TIPS glue embolization and were reviewed met the exclusion criteria and were thus excluded due to sample size < 5, or mixed results from only combined embolic agents (NBCA + mechanical embolic agents), or because no clear endpoints for the injection of NBCA-Lipiodol® were reported. This is the explanation.

  1. As the authors state one of the main weakness of this work, apart from heterogeneity of methods and techniques, regards the variability of clinical situations which has not been investigated.

Reply: Thank you very much for your comment. Clinical situations were indeed not always clearly reported or detailed in the studies included in the analysis. However, we believe that this analysis offers a good overview of NBCA-Lipiodol® mixture injection outcomes in variceal bleeding in real-life conditions. From a statistical point of view, it was impossible to extract data from the studies about the variability of clinical situations.

  1. The manuscript is well written, it is clear and the statistical analysis seems appropriate. No major concerns with publication in the present form.

Reply: Thank you very much for your comment. The manuscript has been improved as suggested.

Reviewer 2 Report

This is an interesting review analyzing the results of NBCA glue injection in the spectrum of variceal gastrointestinal bleeding. The study was well designed and with no methodological issues. However, there were several limitations precluding its publication with no changes. First, all the variceal bleedings, both esophageal and gastric, were analyzedtogether. Consensus conferences and robust literature confirmed that the combination of EVL + vasoactive drugs were the optimal treatment for esophageal and GOV1 varices; just considerin sclerotherapy as a second choice if EVL was not feasible. Taking this into consideration, the authors must: 1) clarify if NBCA glue injection was or not associated to vasoactive drugs in the studies included; and 2) made a separate analysis according to the type of varices treated, that is to say, esophageal and GOV1 varices; or GOV2 and IGV1 varices. 

Moreover, the endpoints were diferent that the ones considered relevant in variceal bleeding (according to previous Baveno Consensus Conferences): 5-day control of bleeding, late rebleeding (from day 5 to day 42), and 6 weeks survival.

In my opinion, this review may be accepted if the previous considerations are taken into account and correctly solved.

Author Response

Response to Reviewer 2

  1. I don't feel qualified to judge about the English language and style.

Reply: Thank you very much for your comment. English language has been checked by a native speaker as suggested.

  1. This is an interesting review analyzing the results of NBCA glue injection in the spectrum of variceal gastrointestinal bleeding. The study was well designed and with no methodological issues. However, there were several limitations precluding its publication with no changes. First, all the variceal bleedings, both esophageal and gastric, were analyzed together. Consensus conferences and robust literature confirmed that the combination of EVL + vasoactive drugs were the optimal treatment for esophageal and GOV1 varices; just considering sclerotherapy as a second choice if EVL was not feasible. Taking this into consideration, the authors must: 1) clarify if NBCA glue injection was or not associated to vasoactive drugs in the studies included; and 2) made a separate analysis according to the type of varices treated, that is to say, esophageal and GOV1 varices; or GOV2 and IGV1 varices. 

Reply: Thank you very much for your comment. We fully agree. Details regarding the type of varices and the use of vasoactive drugs have been added in Table 1 as suggested. Unfortunately, the use of vasoactive drugs was not reported in many studies as mentioned in Table 1. Separate analysis according to the type of varices treated was also unfortunately not statistically feasible with the data that were reported by the included studies. Almost all studies did not show sub-group results regarding the type of varices but reported only mixed results leading to the inability to extract separate data. This point is very interesting and our meta-analysis reflects the heterogeneity of real-life reported results on this topic despite recommendations, which represents the strength of this meta-analysis in our opinion. It is of utmost importance for readers to be aware of this discrepancy between real-life reported results with this technique and the  recommendations on it, meaning that upcoming studies on this topic should be more rigorous with these criteria. All these points have been described more in detail in the text of the manuscript in both materials and methods, and discussion sections, for more understanding. In addition, all references have been renumbered through the manuscript.

  1. Moreover, the endpoints were different that the ones considered relevant in variceal bleeding (according to previous Baveno Consensus Conferences): 5-day control of bleeding, late rebleeding (from day 5 to day 42), and 6 weeks survival.

Reply: Thank you very much for your comment. We fully agree. However, we could not use the endpoints recommended by the Baveno consensus in our meta-analaysis. Indeed, considerable variability was found regarding the studies endpoints. A significant proportion of the included studies were published before the Baveno recommended endpoints have been fully accepted. Furthermore, these recommendations have evolved during the last 20 years. It could explain the used endpoints in these studies. Last, many of these studies were retrospective. Maybe missing data did not allow the use of the Baveno criteria in most of them. For all these reasons and because of this heterogeneity in reported results, we decided to use other endpoints, especially more conventional endpoints, extractable more homogeneously from a statistical point of view. Again and as mentioned in previous remark, this point is very interesting and our meta-analysis reflects the heterogeneity of real-life reported results on this topic despite recommendations, which represents the strength of this meta-analysis in our opinion. It is of utmost importance for readers to be aware of this discrepancy between real-life reported results with this technique and the  recommendations on it, meaning that upcoming studies on this topic should be more rigorous with these criteria. We discussed this limitation in the last paragraph of the discussion section for more understanding.

  1. In my opinion, this review may be accepted if the previous considerations are taken into account and correctly solved.

Reply: Thank you very much for your comment. The manuscript has been improved as suggested.

Round 2

Reviewer 2 Report

The autors correctly answered all the issues previously identified as problems and also provided the modification in the article text corresponding to the clarification required. I think the article can be accepted as it is now.

Author Response

  1. I don't feel qualified to judge about the English language and style.

Reply: Thank you very much for your comment. English language has already been checked by a native speaker as suggested.

  1. The authors correctly answered all the issues previously identified as problems and also provided the modification in the article text corresponding to the clarification required. I think the article can be accepted as it is now.

Reply: Thank you very much for your comment. Everything has been solved as suggested. No further revision has been made.